# The Gut Microbiota Involvement in the Panorama of Muscular Dystrophy Pathogenesis

**DOI:** 10.3390/ijms252011310

**Published:** 2024-10-21

**Authors:** Cristina Russo, Sofia Surdo, Maria Stella Valle, Lucia Malaguarnera

**Affiliations:** 1Section of Pathology, Department of Biomedical and Biotechnological Sciences, School of Medicine, University of Catania, 95123 Catania, Italy; 2Italian Center for the Study of Osteopathy (CSDOI), 95124 Catania, Italy; s.surdo98@yahoo.com; 3Section of Physiology, Department of Biomedical and Biotechnological Sciences, University of Catania, 95123 Catania, Italy; m.valle@unict.it

**Keywords:** muscular dystrophies, intestinal microbiota, gut–muscle axis, muscular immunity, intestinal microbiota-derived metabolites, mitochondrial muscle biogenesis

## Abstract

Muscular dystrophies (MDs) are genetically heterogeneous diseases characterized by primary skeletal muscle atrophy. The collapse of muscle structure and irreversible degeneration of tissues promote the occurrence of comorbidities, including cardiomyopathy and respiratory failure. Mitochondrial dysfunction leads to inflammation, fibrosis, and adipogenic cellular infiltrates that exacerbate the symptomatology of MD patients. Gastrointestinal disorders and metabolic anomalies are common in MD patients and may be determined by the interaction between the intestine and its microbiota. Therefore, the gut–muscle axis is one of the actors involved in the spread of inflammatory signals to all muscles. In this review, we aim to examine in depth how intestinal dysbiosis can modulate the metabolic state, the immune response, and mitochondrial biogenesis in the course and progression of the most investigated MDs such as Duchenne Muscular Dystrophy (DMD) and Myotonic Dystrophy (MD1), to better identify gut microbiota metabolites working as therapeutic adjuvants to improve symptoms of MD.

## 1. Introduction

Muscular dystrophies (MD) consist of a variety of disorders characterized by primary skeletal muscle atrophy. MDs are genetically heterogeneous diseases mostly originating from mutations in genes encoding extracellular, membrane-associated, and intracellular proteins connecting the cytoskeleton to the basal lamina [1]. Several studies have indicated the intricate nature of MD. Mutations that affect the integrity of this complex produce a fragile sarcolemma and are the main feature of MD [2]. The pathological expression of MD is strictly linked to the types of mutation, the absence of the dystrophin-glycoprotein complex (DGC), and the degree of respiratory and cardiac involvement [3]. Immunoresponse, mitochondrial dysfunction, and metabolic anomalies lead to inflammation, fibrosis, and adipogenic cellular infiltrates. Overall, these phenomena are important aspects of MD, suggesting an environmental cause of chronic muscle inflammation. Moreover, dystrophic patients suffer from gastrointestinal disorders, dysphagia changes in motility, and smooth muscle fibrosis. Manometry shows low or no pharyngeal contractions and decreased pressure of the upper esophageal sphincter [4].

The gut microbiota plays a substantial role in shaping and regulating immune responses. Alterations in the intestinal microbial composition, known as dysbiosis, have been linked to various autoimmune and immune-mediated diseases [5]. The gut microbiota also plays an important role in muscle homeostasis [6]. 

Recent studies on murine models with muscular dystrophy have highlighted the role of the intestinal microbiota in MD. However, it is not yet fully understood how dysbiosis can affect the pathophysiology of MD. The purpose of this review is to examine whether resident intestinal microbial communities are implicated in the progression of muscular dystrophies such as Duchenne Muscular Dystrophy (DMD) and Myotonic Dystrophy (MD1), and how they influence clinical and phenotypic variability in dystrophic patients by modulating metabolic and immune responses in order to offer strategies for intervention on the metabolites of the intestinal microbiota to manage or recover MD symptoms.

## 2. Gut–Muscle Axis: Gut Microbiota and Skeletal Muscle Metabolism

The gut microbiota consists of 10^14^ bacteria contained in the digestive tract and classified into numerous species, families, and phyla. 

The microbiota can be influenced by intrinsic factors such as the host’s genetic background, age, race/ethnicity, and sex and by extrinsic factors such as diet, drugs, physical activity, chronic stress, smoking, alcohol consumption, time variation, hygiene conditions, and environmental conditions. Particularly, bioactive compounds present in nutrients influence the human microbiota and significantly affect the health of the host. The gut microbiota is involved in a plethora of host physiological processes and can be regarded as a “metabolic organ” [7]. It is crucial for the homeostatic balance of metabolism and host health. Due to functional crosstalk with other organs such as the brain, heart, liver, adipose tissue, and skeletal muscle, disturbances in the composition and function of the intestinal microbiota such as dysbiosis have been associated with many diseases, such as brain disorders (depression, autism, Alzheimer’s disease, sarcopenia) and metabolic disorders (obesity, type 2 diabetes, insulin resistance). Interactions between gut microbiota and skeletal muscle have recently become an important field of investigation. Skeletal muscle changes its physiology or is affected by pathological conditions in response to the gut microbiota composition. The gut microbiota may influence skeletal muscle through different mechanisms including disruption of the mucosal barrier system, regulation of immune responses, oxidative stress, mitochondrial function, regulation of nutrient absorption from food and the production of microbial metabolites including branched chain amino acids, secondary bile acids (SBAs) and short-chain fatty acids (SCFAs) such as acetate, butyrate, and propionate [8,9] (Figure 1).

SCFAs produced by gut microorganisms through fermenting dietary fiber or protein, are important in protecting the gut barrier. SCFAs are a preferred energy resource for colonocytes and facilitate the expression of tight junction proteins, such as claudin, occludin and zonulin, which control gut barrier permeability. Their action depends on activating G-protein coupled receptors and inhibiting histone deacetylase (HDACs). A deficiency in the of colonic SCFAs production, due to the reduction of the relative abundance of SCFA-producing gut microorganisms, causes a downregulation of colonic CEA cell adhesion molecule 1 (CEACAM1) expression by increasing HDAC3 levels, resulting in impaired apical epithelial integrity of the colon [10].

Metabolites can cross the gut barrier and, either directly or after being transformation into co-metabolites by the host, act on skeletal muscle metabolism. Therefore, muscle dystrophies could be affected by a significantly disturbed metabolism of skeletal muscle.

The impairment of the gut barrier promotes the translocation of bacteria and their components, such as LPS, into the systemic circulation, increasing the triggering of inflammatory responses that can have effects throughout the body. Several circulating pro-inflammatory mediators, activated by alterations in the composition of the intestinal microbiota, have recognized effects on skeletal muscle.

Moreover, the exhaustion of intestinal bacteria by treatment with broad-spectrum antibiotics leads to a decrease in skeletal muscle resistance and an alteration of glucose homeostasis, as indicated by the reduction of SCFA expression and glucose transporters in the ileum and the reduced glycogen content in muscle [11] (Table 1). 

The transplantation of uncultured and intact human fecal samples from older adult humans into germ-free mice to examine the causal role of the gut microbiota on the percentage of lean body mass and physical function showed that family level *Prevotellaceae*, genus level *Barnesiella* and *Prevotella*, and species level *Barnesiella* may be involved in mechanisms related to the maintenance of muscle strength in older adults [12] Comparing the skeletal muscle of germ-free mice in which the intestinal microbiota was absent with the skeletal muscle of pathogen-free mice that had a gut microbiota, it was found that the germ-free skeletal muscle of the mouse showed reduced muscle mass, decreased expression of insulin-like growth factor 1 and reduced transcription of genes associated with skeletal muscle growth and atrophy compared to control mice [9]. Furthermore, germ-free mice showed several variations in the levels of amino acids such as glycine and alanine, and reduced serum choline, the precursor of acetylcholine, an important neurotransmitter carrying signals between muscle and nerves at neuromuscular junctions. At the molecular level, the skeletal muscle of germ-free mice showed a reduced expression of neuromuscular junction assembly genes such as Rapsyn, a 43 kDa receptor-associated synaptic protein, and low-density lipoprotein receptor-related protein 4 (Lrp4), compared to control mice [9]. However, it was shown that transplantation of the gut microbiota in germ-free mice induces an increase in skeletal muscle mass, and a reduction in markers of muscle atrophy [9]. In other studies, it was revealed that following the gut microbiota transfer from obese or lean pigs to germ-free mice, they reproduced the skeletal muscle fiber texture of the donors [24]. 

## 3. Gut Microbiota-Derived Metabolites and Muscle Mitochondrial Biogenesis

The muscle–gut axis is often generated by the interaction between microbiota and mitochondria [8]. Mitochondria being an important organelle for energy metabolism and a source of reactive oxygen species (ROS) are able to influence skeletal muscle function [19]. Skeletal muscle regulates crucial functions such as glucose absorption, fatty acid oxidation and protein metabolism [20]. Evidence indicates that the gut microbiota may affect the mitochondria resident in muscles, and through ROS, reactive nitrogen species (RNS) and various cytokines such as IL-6 cause alterations in innate immunity [13]. In germ-free mice skeletal muscle, a reduced transcription of genes associated with mitochondrial function has been found. Transplantation of the intestinal microbiota from pathogen-free mice to germ-free mice resulted in an improvement of the oxidative metabolic capacity of muscle [9]. Remarkably, gut microbiota-derived metabolites, such as SBAs, SCFAs, and hydrogen sulphide (H_2_S), directly or indirectly promote ROS formation and mitochondrial energy metabolism through adenosine monophosphate kinase (AMPK), peroxisome proliferator-activated receptor γ coactivator 1α (PGC-1α), and sirtuin 1 (SIRT1) pathways [14,15]. In particular, AMPK induces glucose uptake and fatty acid oxidation and promotes glycolysis if there is a depletion of cellular energy [16]. PCG-1α regulates mitochondrial biogenesis and oxidative metabolism by facilitating fiber-type switching from glycolysis to oxidation [17]. SIRT1, induced by SBAs, acts as an energy receptor to redox reactions and may promote mitochondrial biogenesis through coactivator-1α γ receptor activity activated by PGC-1α deacetylation [13]. PGC-1α is upregulated through SIRT1 activation to promote myofibril fiber switch, downregulating FOXO3, which promotes mitochondrial biogenesis, and downregulates MURF-1 and Atrogin-1 expression. SCFA stimulates glycogen uptake and ATP release and improves skeletal muscle function. It has been observed that in mouse models fed a high-fat diet, butyrate supplementation is associated with improved insulin sensitivity, increased PGC-1α and protein kinase activated by AMP (regulation of energy metabolism) and a higher percentage of type 1 fiber in skeletal muscle [25]. In addition, SCFA muscle-receptors specific FFAR3 and FFAR2 promote insulin sensitivity and modulate glucose absorption. In intestinal cells SBAs operate indirectly by activating farnesoid X-receptor (FXR) and, thus, are able to induce fibroblast growth factor 19 (FGF19)/fibroblast growth factor 15 (FGF15). FXR activation reduces insulin resistance and protects against muscle fat deposition [26]. Activation of FXR also reduces the steroid-1c response protein (SREBP-1c), the carbohydrate response protein (ChREBP) and the peroxisome proliferator receptor α (PPAR-α), which are all found in skeletal muscle that plays a role in fatty acid synthesis [27], muscle fiber type determination and fatty acid absorption and oxidation [27]. Disruption of FXR signaling leads to atrophy through disruption of fibroblast growth factor 19 (FGF19; FGF15 in rodents) and downstream extracellular signal (ERK) regulated protein kinase signaling pathways (ERK) [28] (Figure 1). On the other hand, mitochondria can influence the composition and function of the gut microbiota by regulating intestinal mucosal barrier function and immune responses [29].

## 4. Interaction between Skeletal Muscle and the Immune System

Recent advances have highlighted the interaction between skeletal muscle and the immune system. Leukocytes are well-represented immune cells in healthy skeletal muscle (total muscle contains 4 × 10^10^ leukocytes) [30]. Intramuscular leukocytes comprise various cell types, including neutrophils, eosinophils, CD8 cytotoxic T cells and regulatory T cells (Treg). Mostly intramuscular leukocytes are monocytes/macrophages largely placed in the sheath of connective tissue surrounding whole muscles or near blood vessels [31]. Like satellite cells, resident macrophages reside in a quiescent state in healthy muscle, however, increased muscle use or trauma causes them to be rapidly activated, necessary for normal muscle regeneration [30]. Macrophages, eclectic immune cells, can undertake pro-inflammatory (M1) or anti-inflammatory (M2) functions depending on environmental signals [32]. While the phenotype M1 reflects macrophagical activation by pro-inflammatory T helper 1 (Th 1) amplifying inflammatory responses. M2 macrophages, are activated by Th2 and are associated with inflammation resolution and tissue repair [32]. Muscle injury due to intensive exercise, trauma, freezing, burns, toxins, or some diseases trigger an inflammatory response and a rapid increase of more than 100-fold of intramuscular leukocytes. Resident macrophages, by releasing the neutrophil chemoattractors CXC-chemokines ligand 1 (CXCL1) and CC-chemokines ligand 2 (CCL2), promote a massive neutrophil invasion in the damaged muscle [33]. This neutrophil invasion is an important response to acute muscle injury that influences the state of activation of succeeding immune cell populations. Neutrophils are recruited to the site of skeletal muscle damage primarily through interleukin-6 (IL-6) and IL-8. This stimulates the recruitment of macrophages to skeletal muscle and the activation of various inflammatory pathways, including the Janus kinase signal transducer (JAK) and transcription activator (STAT), mitogen-activated protein kinase (MAPK) and the activated B cell nuclear kappa-light-chain-enhancer (NF-κB). After the neutrophil invasion, the circulating monocytes/macrophages invade a muscle environment packed with pro-inflammatory cytokines such as interferon-γ (IFNγ) and tumor necrosis factor-alpha (TNF-α) [34,35]. The initial inflammatory response and early myogenesis, when satellite cells are activated and begin to proliferate and differentiate, are two co-regulated processes. In fact, both IFNγ and TNF-α coordinates the initial inflammatory response with the early stages of regeneration. Increased levels of IFNγ and TNF-α expression in damaged tissue concur with the increase in neutrophils, macrophages and activated satellite cells expressing MYOD [35,36]. 

Both these cytokines can polarize macrophages in the pro-inflammatory phenotype M1. In the post-lesion stages, it is crucial to dissect the roles of macrophage subpopulations. The inflammatory infiltrate may contain a mix of M1 promoted tissue injury/inflammation and M2 polarized macrophage phenotypes that show anti-inflammatory action including reparative fibrosis [37]. M2 macrophages promote the regeneration of skeletal muscle tissue through the secretion of anti-inflammatory cytokines. Lastly, T cells proliferate and in particular, T regulatory lymphocytes (Tregs) are recruited to the regeneration area and signal to satellite cells to proliferate and differentiate (Figure 2). Therefore, the regenerative process in skeletal muscle is highly regulated and coordinated by macrophages M1 and M2 [38]. Both M1 and M2 macrophages are crucial for proper signaling to satellite cells after skeletal muscle damage to start and end the repair process. M1 macrophage permanence for a longer period (>2 days), can defer the recovery and amplify the inflammatory response [39]. M1 macrophages interact with satellite cells, starting their migration and proliferation and promoting myofibers growth [40]. On the other hand, M2 macrophages start the process of differentiation of satellite cells in muscle [40]. M2 macrophage premature arrival or longer permanence can cause alteration and dysregulation in healing tissue and control of the regenerative process by satellite cells [41].

## 5. Gut Microbiota Effects on Muscle Immunity

The gut microbiota is important for the maturation and differentiation of immune cells, for a suitable host immune response [21]. The interactions between the microbiota and intestinal immune cells are firmly and hierarchically regulated, thus their dysfunctions generate serious complications such as chronic inflammatory status, facilitating altered homeostasis even in distant sites including skeletal muscles [42]. In the gut, in the steady state, colon macrophages show an anti-inflammatory phenotype similar to M2 macrophages. M2 are the largest producer of IL-10 that prevents differentiation of M1 macrophages [21]. An important antimicrobial pathway in monocytes and macrophages includes the activation of TLRs. TLR regulation activates immune responses [21]. 

The immune system is equipped with innate immune sensors, Toll-like receptors (TLR), expressed by macrophages, lymphocytes, and dendritic cells (DC) [43]. TLRs stimulate innate intestinal immunity by detecting a broad spectrum of microbial-associated molecular models (MAMP), which include various bacterial antigens such as flagellin, LPS, muramic acid and non-methylated bacterial DNA and peptidoglycan components [44]. Consequently, TLRs trigger a sophisticated cascade of signals, producing the release of NF-κ light-chain-enhancers from activated B cells, which in turn stimulate a large number of genes encoding the humoral immune response such as acute phase proteins, cytokines, chemokines, and antibacterial products [45]. Compromised gut barriers can allow endotoxin translocation, worsening systemic inflammation and promoting skeletal muscle atrophy [46]. In this context, the protective role of butyrate in gut barrier function is crucial as reported by a recent study [18] in which it was shown that butyrate’s gut barrier protection is achieved by inhibiting endotoxin translocation, reducing inflammation, and preventing skeletal muscle atrophy. This investigation demonstrated that butyrate regulates macrophage activities inhibiting LPS-induced M1 polarization and enhancing IL-4-induced M2 polarization. In addition, it was shown that M1 and M2 macrophages affect muscle cell Akt/mTOR/Foxo3a and Fbox32/Trim63 pathways. The Akt/mTOR/Foxo3a pathway is an important regulatory pathway for protein metabolism and turnover [47]. In particular, AKT promotes protein synthesis by phosphorylating mTOR. Moreover, Akt phosphorylates Foxo3a, leading to its nuclear exclusion and, thus, lowering transcription levels of Fbox32/Trim63 ubiquitin ligases induces muscle degradation [48]. Therefore, M1 and M2 macrophages alleviate oxidative stress and contribute to regulating muscle cell atrophy and hypertrophy underscoring the implication of the butyrate-macrophage regulatory pathway in skeletal muscle function [18] (Figure 2).

The gut microbiota and its metabolites also act on NOD-like receptors (NLRs). NLRs are multiprotein cytoplasmic complexes consisting of one of the NLR proteins, such as NLRP1, NLRP3, or NLRC4, acting as sensors of exogenous or endogenous stress or molecular patterns associated with damage. The NLRP3 inflammasome, when stimulated, produces pro-inflammatory cytokines IL-1β and IL-18, which can adversely affect skeletal muscle. NLRs perceive numerous specific microbial molecules and activate the assembly of inflammasomes [49]. Following the detection of the appropriate signal, they are collected together with the adaptor protein, the speck-like protein (ASC) associated with apoptosis, in a multiprotein complex that manages the activation of caspase-1 and the subsequent cleavage of pro-effector inflammatory cytokines including IL-1, IL-6, IL-12, IL-18 and IL-23, which then regulate the balance of Th17/Treg cells towards Th17 cells [50]. For instance, NLPRP6 deficiency has been associated with reduced levels of IL-18, altered immune response, dysbiosis and intestinal hyperplasia [50]. The gastrointestinal microbiota influences both neutrophil migration and function, as well as the differentiation of T cell populations into different types of helper cells (Th), such as Th1, Th2, and Th17 or Treg [21]. Treg cells residing in healthy muscles are able to regulate immune response and directly influence muscular differentiation. They are perfectly placed to act as coordinators for interactions between immune and muscle cells during regeneration and can provide immune memory to muscle. The migration and maintenance of Tregs in the muscle depend on two mechanisms. The first is under ATP/P2X axis control. The other is an effect mediated by IL-33, which is an alarmin generated by fibro adipogenic progenitors (FAPs) and skeletal muscle stem cells that can bind to the IL-1 receptor-like 1 protein present on Tregs, macrophages and FAPs. Therefore, IL-33 is crucial for Treg accumulation within the muscle, and consequently, muscle regeneration [51] (Figure 2). The modulation of the intestinal microbiota alters the regulatory molecules secreted by skeletal muscles and adipose tissues such as myokines and adipokines, whose function is closely dependent on the production of SCFA and branched chain amino acids. In muscle tissue, dysbiosis interferes with the proper development of muscle progenitor cells, most likely through the generation of reactive oxygen species and antioxidant genes.

## 6. DMD

DMD is a progressive wasting disease of skeletal and cardiac muscles. Among MDs, DMD is the most frequent and destructive. It is a hereditary recessive genetic disease with an annual occurrence of ~1:5000 individuals [52]. Mutations in the DMD gene affect the correct synthesis of the dystrophin protein associated with the membrane.

A lack of dystrophin prevents muscle contraction and causes instability of the plasma membrane, resulting in uninterrupted cycles of necrosis and muscle repair. The incessant cycles of degeneration/regeneration in the dystrophic muscles generate persistent muscle deterioration and disruption of the regenerative potential caused by the depletion of satellite cells, this leads to muscular necrosis without healing and inflammation. Moreover, the deterioration of muscle cell membranes due to mechanical stress during contraction/relaxation cycles promotes microlesions [53]. This scenario is exacerbated by variations in metabolism and high energy requests leading to inflammation fibrous remodeling and finally adipose replacement [54]. Fibrosis is a pathological process that contributes to weakness and, in the case of skeletal muscle, impairment of regeneration.

Inflammatory mediators and particular lymphocyte subgroups have been detected in the blood/muscles of DMD patients. In the early stages, due to the regenerative capabilities of muscle fibers, the microlesions may be associated with pseudo-muscle hypertrophy, especially in postural muscles such as the sural triceps [55]. Secondary physio-pathological processes including altered calcium homeostasis, oxidative stress, immunological and inflammatory responses, apoptosis, defective autophagy, and failures in mitochondrial number and function, exacerbate muscle pathology in DMD [56,57]. The most affected skeletal muscles are firstly the locomotor muscles, then the trunk muscles, and lastly the respiratory muscles and the heart, leading to tetraplegia and cardio-respiratory difficulties [58]. DMD affects tissues other than skeletal muscle because dystrophin is expressed ubiquitously throughout the body.

The disruption of relationships between ion channels and components of the dystrophin-glycoprotein complex, and the consequential failure of calcium channels of the transitory receptor potential (TRPC) and cardiac L [59], as well as potassium and sodium channels promote the occurrence of cardiomyopathy [60]. DMD patients also have gastrointestinal alterations including changes in gastrointestinal motility, aerophagia, gastroesophageal reflux, acute dilation, pseudo-obstruction, and constipation, affecting almost 70% of patients as well as life-threatening and metabolic acidosis. These disorders can further lead to insufficient fluid and calorie intake [61], which may contribute to intestinal microbial dysbiosis, further contributing to dystrophy pathology.

## 7. Disturbances of Immune Response in DMD

DMD patients show an increase in the percentage of circulating CD4^+^ and CD8^+^ T lymphocytes expressing high levels of CD49d, encoded by the gene integrin alpha 4. The presence of T lymphocytes CD49d + CD4 + CD3+ and CD49d + CD8 + CD3+ combined with the fibronectin-containing network of the injured muscle indicate a relationship with a fast disease progression as displaying greater trans-endothelial and fibronectin-driven migratory reactions than healthy people [62] (Figure 3). In addition, T cells identified within the DMD muscle tissue principally express TCR belonging to the Vβ2 family [63]. Moreover, DMD muscle fibers are invaded by CD8^+^ T cells [64]. Once activated, cytotoxic CD8^+^ T lymphocytes migrate and recognize specific peptides on the surface of muscle fibers triggering the release of death-related molecules such as granzyme, perforin and TNF-α, amplifying tissue injury [65]. T cells from DMD patients stick more vigorously to myotubes than T cells from healthy people [62].

In dystrophic mice, persistently activated intracellular signaling pathways cause dysregulation of macrophages, triggering disproportionate tissue destruction and the recruitment of CD4/CD8+ T-cells [63]. 

Commensal microbiota influences TLR expression in intestinal epithelial cells and changes in the microbiota may alter the immunogenic roles of TLR, thus affecting the pro-inflammatory phenotype of the intestinal mucosa [64]. Through a combination of TLR and IL-10 signaling regulation, interactions with the intestinal microbiota and with intestinal epithelial cells are generated. The anti-inflammatory properties of M2 could be caused by the interaction with bacterial products. For example, high concentrations in the colon of butyrate, which is produced mainly by the phyla bacteria Bacteroidetes and Firmicutes, moderate the production of proinflammatory factors including IL-6, IL-12 and NOS2 produced by colon macrophages [66]. TLR4 is up-regulated in mdx muscles, a mouse model of DMD its ablation improves the dystrophic phenotype [67]. Chronic damage causes variations of macrophage phenotypes thus the normal regulation of connective tissue production is interrupted. M1 macrophages contribute to oxidative stress and muscle fiber lysis through the production of iNOS-derived NO and promote inflammation and myoblast proliferation by producing Th1 cytokines [68]. Elevated IL-10 production shifts the macrophage population toward an M2c phenotype and deactivates M1 macrophages, causing reductions in iNOS expression and attenuating muscle damage [69]. The switch of macrophages prevents apoptosis of FAP cells, leading to an increase in the deposition of connective tissue. IL-10 can influence the differentiation of myogenic cells [69]. In addition, M2 macrophages in the advanced stages of mdx dystrophy shift to a highly fibrogenic state where there is increased activity of ARG1 driving pathological fibrosis in the dystrophic muscle. During acute muscle damage and the subsequent regenerative phase, IFNγ can induce macrophage polarization in mdx mice muscles.

The muscle of DMD patients shows high expression of cytokines such as IFN-γ, transforming growth factor (TGF)-β and chemokines, such as CCL2, CXCL12 and CXCL14 [70]. Deletion of IFNγ in mdx mice reduces iNOS protein levels in muscle macrophages, and increases in vivo transcription expression for interleukin-4 (IL-4), arginase 1 (ARG1), monoglyceride lipase 2 (MGL2) and FIZZ1 (also identified as resistin-like-α) displaying M2 activation, moving macrophages into a CD206 M2 phenotype [71]. In DMD, the presence and action of DCs is also notable. It is likely that DCs participate in DMD progression. Using their TLR 7 expression they induce cytokine release, extending and enhancing the cycle of inflammation/degeneration/regeneration [72]. DCs cooperate with myoblasts, increasing their proliferation and migration, and cytokine production, [73] such as TGF-β, which has been found robustly induced in DMD patients inducing continuous remodeling of connective tissue and, finally, fibrosis [56]. Moreover, in DMD, the absence of dystrophin has an impact on muscle cell injury generating the release of intramuscular antigens identified by B and T lymphocytes [42]. Tregs also play an important role in the pathogenesis of DMD. IL-10 deficiency, which regulates the activation of Tregs in dystrophic mice, intensifies muscle damage and diminishes strength [74]. In mdx mice antigen specific Treg cells stimulate a more rapid regeneration when compared to polyclonal Treg cells [75]. Likewise, muscle injury and inflammation, worsened by Treg cell depletion, are alleviated by the proliferation of Treg cells, being [76].

## 8. Instabilities of Mitochondrial Function in DMD

Altered metabolism in muscular dystrophy is linked with defective regulation of metabolic pathways, impaired calcium consumption, accumulation of reactive oxygen species, mitochondrial dysfunction, reduced ATP levels, enhanced phosphorylation of the AMP-activated protein kinase [77,78]. As previously mentioned, microbial metabolites can exert a direct influence on oxidative stress. “Gut redox” represents an articulate redox equilibrium within the gastrointestinal tract. Involving a dynamic interaction of various reactive species and antioxidants contribute to maintain redox stability it is crucial in affecting gut health through the composition of the gut microbiota, which contributes to the global homeostasis of the gut immune system. A disorder in the complex stability of gut redox can lead to oxidative stress, marked by an overabundance of ROS and RNS, potentially causing damage to cellular components. Although the gut is primarily anaerobic due to oxygen consumption by bacteria, certain gut cells like epithelial and immune cells such as macrophages and neutrophils, can produce free radicals. The impact of free radicals-associated signaling within the GI tract is closely linked to the local concentration of ROS, leading to damage of crucial cellular components, including lipids, proteins, and DNA, thus contributing to cell dysfunction and premature cell death. Mitochondria are the principal source of ROS. Excessive ROS formation can initiate uncontrollable chain reactions, triggering systemic inflammatory response. The substrate restriction and reduced rate of biochemical reactions of the aerobic metabolism in slow muscles and anaerobic metabolism in fast muscles is the result of reduction of the oxidative consumption of both glucose and free fatty acids [79]. DMD is predominantly linked to sarcolemmal instability and inflammation. Dystrophin also works to anchor neuronal nitric oxide synthase (nNOS) to the sarcolemma. nNOS is the enzyme that produces a diffusible nitric oxide (NO) signaling molecule. In DMD, nNOS is delocalized into the cytosol, resulting in microcirculation alteration and functional ischemia [80]. The delocalization of nNOS in the cytosol causes an increase in reactive nitrogen species, with further adverse effects downstream [81]. In DMD, free radical damage is significantly elevated. In DMD, the microtubule reticulum is thicker and disorganized, resulting in increasing elongation and activation of NADPH oxidase 2 (NOX2), and generation of high levels of ROS [82]. Glutathione levels, which are an important antioxidant, decrease significantly impairing the muscle’s capability to reduce oxidative stress. Moreover, free radical levels increase through dysfunctional mitochondria and inflammatory response [45]. Both cells and tissues of DMD patients display mitochondrial deficits that contribute to muscle weakness. Therefore, DMD can also be regarded as a metabolic disease. In fact, different metabolic pathways are malfunctioning, producing a decline in ATP production [83]. In DMD, due to ADP’s failure to stimulate cellular respiration, mitochondria operate in stress conditions [84]. Interestingly, it has been found that the compromised ATP production in dystrophic mouse mitochondria depends on an intrinsic deficit in complex-I-driven respiration [85]. In addition, decreased levels of mitochondrial aconitase (ACO_2_) disrupt the citric acid cycle. Whereas fatty acid synthase, directing lipid synthesis, is upregulated, sustaining the accumulation of fatty acid [86] (Figure 3). A recent study identified two temporally diverse phases of mitochondrial damage. The early stages consist of depletion of mitochondrial mass followed by an accumulation of dysfunctional mitochondria, resulting in different oxidative fiber patterns in mdx mice. In addition, a progressive mitochondrial biogenesis failure related to augmented deacetylation of PGC-1α promoter was detected [87].

## 9. Composition and Function of the Gut Microbiota in DMD

Gastrointestinal alterations in DMD patients suggest that the composition and function of the gut microbiota in DMD could be changed, with possible effects on the health and myopathy of the host. Variations in the composition of the intestinal bacterial community in DMD can be associated with predisposing factors including a sedentary lifestyle, treatments such as glucocorticoids, or antibiotics for the treatment of respiratory infections [88] could promote gut dysbiosis, exacerbating inflammation and impacting gut barrier functions in treated patients. In addition, the associated patterns of malnutrition would further aggravate the dysbiosis and gastrointestinal dysfunction in dystrophic patients. However, the intestinal microbiota dysbiosis associated with DMD may be only an epiphenomenon of the disease and not a determining factor leading to the loss of muscle function.

Some investigations conducted in mdx mice, reported that they show an increase in intestinal peristalsis related to significant anomalies of the epithelial morphology of the mucosa such as larger villi, reduction of the muscular and submucosal layer, habitually associated with the inflammatory state and the production of NO [89]. Moreover, mdx mice show reduced fecal emissions [90] particularly because of an alteration of the motor complex responsible for migrant mobility in the inner digestive period [91]. Although dystrophin is expressed in smooth muscles, there is still little information about intestinal smooth muscles in DMD [92]. Mdx mice have been also used to assess whether dystrophin deficiency affects the specific composition of the intestinal microbiota and to analyze the intestinal structure and function as well as expression of genes related to bacterial-derived metabolites in the ileum, in blood and skeletal muscles and to study inter-organ interactions. Comparing mdx mice with B10 wild-type underscored a distinctive intestinal bacterial composition in mdx mice linked to modifications of specific plasma and muscle inflammatory biomarker levels. Mdx mice showed a significant reduction in the total number of different operational taxonomic units and their abundance. The taxonomic modification involved phyla of *Actinobacteria*, *Proteobacteria*, *Tenericutes*, and *Deferribacteres* and the included genera. In particular, the Mucispirillum genus and Deferribacteraceae family in the Deferribacteres phylum; the *Enterorhabdus* genus and *Coriobacteriaceae* family in the Actinobacteria phylum; and the *Rhdospirillaceae* family in the Proteobacteria phylum. Remarkably, the Deferribacteres phylum and related taxa were only detected in *mdx* mice and not in wild-type mice, suggesting a specific phylum and taxa related to dystrophin deficiency [93]. In addition, in *mdx* mice a significant presence of LPS producing Gram-negative bacteria (Deferribacteres phylum, *Bacteroides* genus) was found, with reduced intestinal motility as well as gene expressions of the ileum, which suggested an increased intestinal porosity contributing to low-grade inflammation [93]. Furthermore, intestinal motility and gene expression of tight junction proteins (*Tjp1* and *Tjp2*) essential for intestinal functionality/permeability preservation were reduced in the mdx mice ileum [94]. Tight junctions are essential in regulating transition of nutrients and waste items from the lumen to the bloodstream. In case of impairments of TJs the consequent broken intestinal permeability allows the passage of molecules, toxins, and microorganisms from the intestinal wall into the bloodstream [95]. 

The variation in gut microbiota composition linked to the damage of the intestinal barrier exacerbates inflammation in dystrophin-deficient skeletal muscle, as confirmed by over-regulation of IL-6, TNF-α, and monocyte-1 chemoattractant protein (MCP-1) observed in *mdx* mice and/or bacterial infections in distant organs [93]. In parallel, markers of muscle inflammation such as Toll-like receptor 4 (*Tlr4)*, which recognizes LPS, *Myd88*, *Angptl4* and *Bcat2* (the skeletal muscle-specific isoform of transaminase 2), is augmented in skeletal muscles of *mdx* mice, particularly in the tibialis anterior. In mdx mice intestinal tissues, an increase of IL-6 and TNFα and factor 6 associated with the TNF receptor (TRAF6) that consequently overregulated innate immunity mediators, such as the beta subunit of type 8 beta proteasome (PMSB8), pentraxin-3 (PTX-3), and a viral homolog of viral oncogene V reticuloendotheliosis B (RelB) [96] were found.

Moreover, mass spectrometry for lipid imaging in the tissues of the small mdx mice intestine showed enrichment of phosphatidylcholine (PC) cleavage products such as lysosome-phosphatidylcholines (LysoPC), LysoPC was correlated with an abundance of the bacterial genus *Alistipes* [97]. Remarkably, PC and LysoPC activate various signaling pathways involved in oxidative stress and inflammatory responses triggered by TLR leading to an increased release of cytokines, or IL-1β, IL-6 and TNFα and activation of lymphocytes and M1 macrophages [98,99]. Another investigation performed in mdx mice reported an overexpression of the genera *Alistipes* and *Prevotella*, which positively correlated with splenic cell populations CD44 + CD4 +/CD8+ T cells and Tregs, as well as muscle effector/memory CD44 + CD8+ T cells and central memory CD4+ T cells [96]. Moreover, regulators involved in the modulation of inflammatory events in patients with ulcerative colitis (UC) and Crohn’s disease (CD), such as the IL-33/ST2 axis and IP, are commonly altered in DMD [62] (Figure 3). Therefore, gut microbiota plays a pivotal role in the pathogenesis of DMD. Altered gut microbiota content impairs the innate immune response and the expression of genes involved in early myogenesis, altering muscle metabolism, architecture, and force [96].

## 10. Metabolic Profile in DMD

Dystrophy tissues show progressive metabolic changes due to the disease itself, but also partly due to obesity or metabolic syndrome in the later stage of the disease. The metabolic disorder may be a determinant of disease progression of this hereditary disease. DMD is relatively understudied concerning circulating metabolites and metabolic pathways in DMD patients. Recent evidence has detected several differential metabolites associated with gut microbiota such as phenylacetylglutamine (PAGln), hippuric acid, bile acids, which may be related to nutritional disorders and intestinal muscle dysfunction in DMD patients [77]. The increased plasma PAGln levels impacts in augmented risks of cardiovascular and thrombosis potential disease [97]. Instead, the serum levels of hippuric acid in DMD declined, likely due to the intestinal microbiota disorders in patients with DMD. In addition, in DMD patients, levels of the bile acid glycoursodeoxycholic acid (GUDCA) are elevated. Interestingly, deoxycholic acid destroying cholic acid-enterohepatic circulation promotes dysregulation of intestinal inflammation. GUDCA is an antagonist of the bile acid receptor that increases hepatic bile acid levels. Deoxycholic acid is associated to disarrays in bile acid metabolism and downregulation of the ileal FXR/FGF 15 axis [99]. FXR-FGF15/19 signaling affects skeletal muscle function as a symbiotic regulator of gut microbiota [28]. FGF15/19-dependent ileal FXR signaling facilitates the increase in skeletal muscle protein synthesis, suggesting that it may be a potential therapeutic target for sarcopenia [28]. Therefore, the intestinal FXR-FGF15/19 signaling pathway may play an important role in DMD. Metabolic profile analysis in mdx mice showed alterations in carbohydrate and amino acid metabolism pathways and a significant decrease in the expression of SCFA biosynthetic enzymes and ketone bodies [92; 97]. mdx mice showed a significant reduction in the gene content of key biosynthetic enzymes SCFA such as propionyl-CoA: succinate CoA transferase, propionate CoA-transferase, butyril-CoA: acetate CoA-transferase and butyrate kinase [96].

Ffar2 (SCFA receptor) downregulation was also observed, which emphasized that a variation of the gut microbiota SCFA production and its bioavailability for skeletal muscles could influence its metabolism and contractile function [93]. Likewise, *Bcat2* differential profiles in tibialis (up-regulation) and soleus (down-regulation) indicate a complex response between muscle phenotypes and suggests that variations of the gut microbiota in *mdx* mice could contribute to different bioavailability of branched-chain amino acids [93]. Furthermore, a significant reduction in the concentration of amino acids such as alanine, aspartic acid, methionine, and phenylalanine was observed in these mice. Interestingly, tartrate can act as a muscle toxin by inhibiting the production of malic acid [96].

Furthermore, in mdx mice, adiponectin has been found reduced in the blood and its receptor modulated in muscles. The expression of *Adipor1*, the supposed highest driver of adiponectin cascade in skeletal muscle, influences the different expression on the muscle of mdx mice (reduced in tibialis anteriore, and decreased in soleus). This dissimilarity may be due to a different metabolism related to the type of muscle. In fact, the levels of adiponectin in the skeletal muscle could point to the modulation of the genes of the heavy myosin chain toward the oxidative phenotype [100]. Additionally, a significant trend of reduced levels of bacteria-producing butyrate (such as Roseburia and Ruminococcaceae) has been revealed, correlated with lower butyrate levels in cachexia individuals and mice. Butyrate’s gut barrier integrity protection is accomplished by modulating macrophage polarization, inhibiting endotoxin (LPS) translocation and reducing serum levels of inflammatory cytokines and IL-6. Compromised gut barriers can induce endotoxin translocation, exacerbating systemic inflammation and aggravating skeletal muscle atrophy [18]. The protective role of butyrate in gut barrier function precisely because it is essential in anti-inflammatory processes, immune regulation and homeostasis preservation, the reduction of butyrate-producing bacteria points to a potential mechanistic relationship between the imbalance of the gut ecosystem and the progression of cachexia [101]. Butyrate is also a general inhibitor of histone deacetylase (HDAC). HDACs are enzymes that remove the acetyl group from lysine residues, making DNA less likely to be transcribed [102]. HDACs play a crucial role in the development and maintenance of skeletal muscle (Figure 3). Class II HDACs suppress myoblastic differentiation by regulating myoblast activity through interactions with the transcription factor myocyte enhancer factor 2 (MEFT) [103]. HDAC 4 and 5 contribute to denervation atrophy through activation of atrogènes [104]. In addition, HDAC inhibitors have been shown to reduce fibrosis and improve muscle function in mdx mouse models of DMD [105]. Butyrate also promotes PGC-1α gene expression, which in one study led to a shift of skeletal muscle fibers from glycolic to mitochondrial oxidative rich in mouse models [105].

## 11. Type 1 Myotonic Dystrophy (MD1)

Type 1 myotonic dystrophy (MD1) is a neuromuscular disease featuring progressive muscle weakness, atrophy and myotonia [106]. MD1 is an autosomal dominant disease attributable to an expansion of the CTG repeats triplet. In particular, the transcription of >50 CTG replications in a CUG triplet in the non-coding region 3 of DMPK, the gene encoding a myosin kinase. The CTG replications inhibit the splicing mechanism of the precursor mRNA, producing the formation of a large spectrum of imperfect and non-functional proteins [107]. Depending on their age of onset of these clinical manifestations and repetitions of CTG, patients with MD1 may present five different phenotypes at the initial examination. These phenotypes are classified as congenital, infant, juvenile, adult and late-onset, with the adult phenotype being the most widespread [108]. MD1 is very variable in the severity of symptoms. Clinically it is characterized by a multisystemic involvement with cognitive deficits, cataracts, cardiac conduction abnormalities and endocrine problems, hypothyroidism, and reproductive problems [109]. Gastrointestinal dysfunctions affecting the entire digestive tract, from the pharynx to the anal sphincter, are often present [110]. Gastrointestinal symptoms include fluctuations between diarrhea and constipation, cramps, abdominal pain or swelling [111]. Moreover, because of the slowing of gastrointestinal motility patients experience slow bowel transit time, late gastric emptying, or gastroparesis [112]. Additional gastrointestinal complaints consist of gastric acid reflux, fecal incontinence, gallbladder, and liver complications [111]. The origin of gastrointestinal manifestations in patients with MD1 is probably multifactorial. A correlation between the size of the CTG repetition and gastrointestinal symptoms of abdominal pain and constipation has been suggested [113]. However, a dysbiosis of the intestinal microbiota could have a significant contribution in gastrointestinal complications experienced by patients with DM [114]. Recently, it has been shown that MD1 patients appear to have a change in the Firmicutes/Bacteroidetes (F/B) ratios, which seems to be associated with soft stools [110]. IL6 is found in the serum of patients with MD1, which is related to muscle weakness and limitations of operational capacity [115]. Therefore, the variation of the F/B ratio may influence inflammation in MD1 patients [110]. Another alteration observed in the microbial signature of the intestinal microbiome in MD1 disease was a relatively lower abundance of Actinobacteria phylum, including *Corynebacterium*, *Rothia* and *DNF00809* in the control groups compared to paired MD1 participants. Moreover, in stool samples of MD1 patients a higher relative abundance of *clostridium* CAG-352 from the phylum Firmicutes, a greater relative abundance of *Lachospiraceae* and *Prevotella* and a decrease in the presence of *Lactobacillus* were found. *Lactobacillus* is a beneficial probiotic bacterium inhibiting both the growth and adhesion of pathogenic bacteria and promoting their removal, thus reducing diarrhea, swelling, and cramping [116].

## 12. Strategies to Modulate Gut Microbiota Composition and Function in MDs

The pathways contributing to aberrant muscle remodeling are modulated by intestinal microbial metabolites, making them plausible targets for prebiotic and probiotic supplementation. 

Metabolites derived from interactions between the gut microbiota and host epithelium, as well as between the microbiome and substrates, are key multiform signaling molecules for muscle function, modifying pathways that contribute to skeletal muscle atrophy, making them a plausible target for additional therapy in muscular dystrophy. In patients with DM and dystrophic murine models, several differential metabolites associated with intestinal microbiota and muscular dysfunction have also been identified. SCFAs (acetate, butyrate, and propionate) have been an important target in the clarification of gut microbial signaling. They play a crucial role in the gut–muscle axis. SCFA supplementation improves muscle mass [116]. Nevertheless, in one study, it was observed that although SCFAs prevented atrophy and increased muscle strength in germ-free mice, they were unable to fully recover the muscle phenotype [9]. Urolithin A is a metabolite produced in the colon by gut bacteria from the ingestion of ellagitannin and ellagic acid-rich foods such as pomegranates, raspberries, strawberries, and walnuts. 

Recent investigations demonstrated that urolithin A supplementation is effective in eliminating dysfunctional mitochondria, in improving mitochondrial fatty acid metabolism and markers of skeletal muscle mitochondrial health such as PTEN-induced kinase 1 (PINK1)/Parkin-mediated mitophagy [117] and in stimulating mitochondrial biogenesis regulating effectors, such as the PGC-1α and SIRT1 [118]. Urolithin A also promotes the regenerative capacity of muscle stem cells and increases skeletal muscle breathing capacity, increasing the survival of dystrophied mice [22]. It has been proposed that Urolithin A removing dysfunctional mitochondria, reducing ROS formation and inducing the release of mtDNA and cardiolipins, could exert an anti-inflammatory effect [91]. In addition, urolithin A promotes the reduction of inflammatory markers such as C-reactive protein (CRP) and inflammatory cytokines [23]. Interestingly, have been identified three metabotypes with quantitative and qualitative differences based on dissimilar Urolithin producing profiles (UM-A, UM-B, and UM-0) in populations worldwide [119]. Therefore, it would be interesting individualizes the metabotypes of patients with DMD to personalize nutrition intervention or supplementation.

In addition to the diagnostic importance of microbial composition, there is no data on the levels of TJ proteins in dystrophy-related muscle. TJ proteins such as zonulin that has been identified as a regulator of intestinal TJs and could be used as potential markers. 

Zonulin is one of the proteins acting as regulator of intestinal TJs. Increased permeability is associated with higher zonulin levels. Extracellular zonulin stimulates EGF receptor signalling and protease activated receptor 2 (PAR-2) disassembling the TJs via interaction between actin and TJ proteins [31]. 

The evaluation of dysbiosis in MD could be translated into clinical practice suggesting the intake of probiotic supplements and prescribe diets to reduce malnutrition and gastrointestinal symptoms. 

## 13. Concluding Remarks 

The intestinal microbiota can be considered a transducer of nutrient signals to the host, with the ability to generate pro-anabolic signals and produce mediators that regulate metabolic homeostasis, immune response, inflammation, and mitochondrial biogenesis [112]. Investigations from germless mice demonstrate that gut microbes may be linked to the function and quality of skeletal muscle. The effects of gut microbiota deficiency on skeletal muscle recently detected in animal models, revealed that a lack of gut bacteria causes muscle mass loss [116,120]. A comparison of germ-free mice lacking gut microbiota and pathogen-free mice with gut microbiota revealed skeletal muscle atrophy and decreased muscle mass in germ free mice [9]. The composition of the gut microbial population can have a considerable impact on gastrointestinal symptoms and in worsening the evolution of DM. Growing evidence supports the potential of an additional regimen directed to the microbiota designed to optimize the signaling of the gut–muscle axis, which could alleviate muscular atrophy in MD. *Lactobacillus plantarum* supplementation has shown positive effects on muscle mass and function in both mouse models and humans [113,114,121] alleviating muscle atrophy and reducing the expression of the markers of inflammation and atrophy, which can generate muscular dystrophy [115]. 

Moreover, in a fecal transplantation experiment in which mdx mice were given eubiotic microbiota was observed a reduction in the inflammatory muscle response and an inversion of other deficits in pathology and muscle function such as the decrease in the area of the transverse section of the myofibers, the fibrous infiltrate, the reduction of lipid and carbohydrate metabolism and reduction of tetanus strength in the anterior tibial muscle in mdx mice [120]. The main purpose of this review is to encourage investigations on the impact of gut microbiota on the course and progression of muscular dystrophies. 

We felt it was important to mention the alterations of the microbiota composition of patients with DM1 because the implications of the altered microbial composition present in this pathology may have the same repercussions as shown by the experiments on murine models of DMD. We believe that these alterations may also influence the evolution of other muscular dystrophies, so, the effects deserve investigation. Most of the current literature focusing on our knowledge about the gut–muscle axis is based mainly on animal model studies. 

Based on what is described in this review, it is necessary to deepen our knowledge of the intestinal muscle axis in MD to translate it into clinical practice. It would be interesting to study the effect of prescribing a personalized diet on the regulation of gastrointestinal symptoms, the administration of probiotics and prebiotics or even considering the use of fecal transplantation as an effective strategy to reduce MD complications. 

Given the shortage of clinical trials, knowledge of the impact of microbiota composition on human dystrophies deserves a great deal of attention in order to design effective diagnostic and therapeutic strategies. 

## Figures and Tables

**Figure 1 ijms-25-11310-f001:**
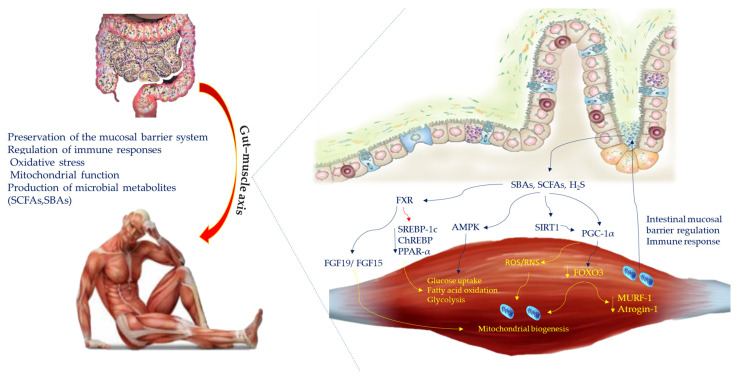
Promoters of muscle–gut axis. The gut microbiota affects skeletal muscle protecting the mucosal barrier system and regulating immune responses, oxidative stress, and mitochondrial function by microbial metabolites production. Metabolites such SBAs, SCFAs, and H_2_S cross the gut barrier, inducing ROS formation and mitochondrial energy metabolism through AMPK/PGC-1/SIRT1 pathways. AMPK induces glucose uptake and fatty acid oxidation and promotes glycolysis. PCG-1α regulates mitochondrial biogenesis and oxidative metabolism. SIRT1 promotes PGC-1α deacetylation and downregulates FOXO3, which promotes mitochondrial biogenesis, and downregulates MURF-1 and Atrogin-1 expression. SBAs activate FXR which in turn FGF19/FGF15. Activation of FXR decreases SREBP-1c, ChREBP and PPAR-α, which induce fatty acid absorption and oxidation. Abbreviations: AMPK, adenosine monophosphate kinase; ChREBP, carbohydrate response protein; FXR, farnesoid x receptor; FGF19/FGF15, fibroblast growth factor 19/15; FOXO3, forkhead box O3; H_2_S, hydrogen sulphide; MURF-1, muscle RING-finger protein-1; PGC-1α, peroxisome proliferator-activated receptor γ coactivator 1alpha; PPAR-α, peroxisome proliferator receptor alpha; ROS, ROS—reactive oxygen species; SBAs; secondary bile acids; SCFAs, short-chain fatty acids. SIRT1, sirtuin 1; SREBP-1c, steroid-1c response protein.

**Figure 2 ijms-25-11310-f002:**
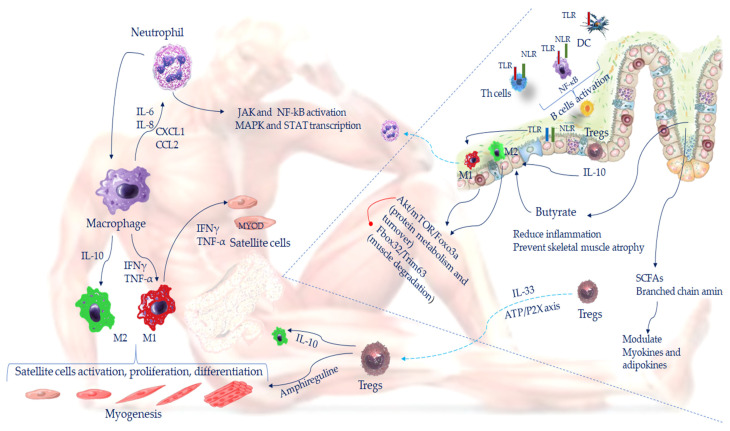
Gut microbiota effects on muscle immunity. Interactions of gut microbiota with immune cells and intestinal epithelial cells are generated by TLRs and NLRs. Gut microbiota influences neutrophil migration and function, as well as M1 and M2 macrophages polarization and T cell differentiation into Th1, Th2, and Th17 or in Treg. M1 and M2 macrophages affect muscle cell Fbox32/Trim63 and Akt/mTOR/Foxo3a pathway, contributing to muscle cell atrophy muscle and hypertrophy regulation. Resident macrophages release the neutrophil chemoattractors CXCL1, CCL2 and IL-6 and IL-8 to stimulate neutrophil and activation of JAK, STAT, MAPK, and NF-κB pathways. Both IFNγ and TNFα induce neutrophils, macrophages and activated satellite cells expressing MYOD. Both M1 and M2 macrophages interact with satellite cells, starting their migration and proliferation and promoting myofibers growth. Tregs migrate in the muscle by mechanisms mediated by ATP/P2X axis and IL-33. Treg cells residing in healthy muscles are able to regulate immune response and directly influence muscular differentiation. Gut microbiota-derived metabolites such as SCFAs and branched chain amino acids control inflammation and molecules secreted by skeletal muscles and adipose tissues such as myokines and adipokines. Particularly, butyrate produced by gut bacteria regulates macrophage activities inhibiting LPS-induced M1 polarization and enhanced IL-4-induced M2 polarization controlling inflammatory response and protecting gut barrier by inhibiting endotoxin translocation and preventing skeletal muscle atrophy. Abbreviations: Akt/mTOR/Foxo3a, *Akt* kinase/mammalian target of rapamycin/forkhead box O3a; ATP/P2X adenosine-triphosphate/Purinergic Receptor2; CCL2, CC-chemokines ligand 2; CXCL1, CXC-chemokines ligand 1; IFNγ, Interferon-γ; IL-6, IL-8, IL-33, interleukin-6, 8, 33; JAK, Janus kinase signal transducer; F-box32/Trim63 F-Box Protein 32/tripartite motif containing 63; MAPK, mitogen activated protein kinase; LPS, MYOD, myogenic determination factor; NF-κB, nuclear factor kappa-light-chain-enhancer of activated B cells; NLRs, NOD-like receptors; STAT, signal transducer and activator of transcription; TNF-α, tumor necrosis factor-alpha.

**Figure 3 ijms-25-11310-f003:**
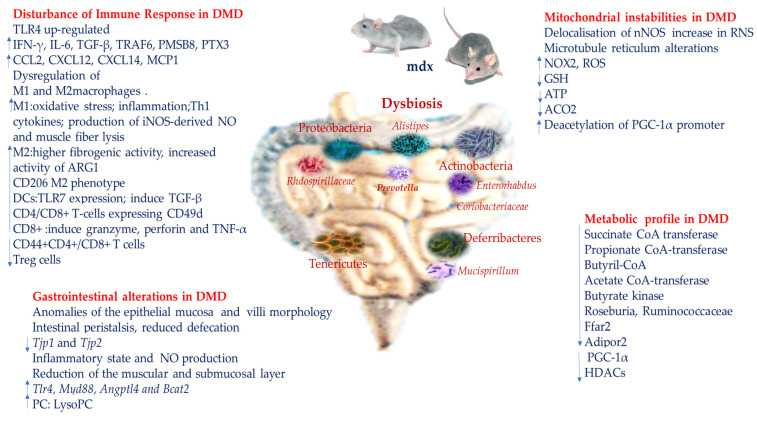
Effects of dysbiosis in DMD. Abbreviations: ACO2, Aconitase 2; Adipor2, Adiponectin receptor 2; ARG1, Arginase 1; ATP, *adenosine-triphosphate*; Bcat, CCL2, CC Motif Chemokine Ligand 2; CD, Cluster of differentiation; CXCL, C-X-C Motif Chemokine Ligand; Ffar2, Free fatty acid receptor 2; GSH, Glutathione; HDACs, Histone deacetylases; IFN-γ, Interferon gamma; IL-6, Interleukin 6; iNOS, Inducible nitric oxide synthase; MCP1, Monocyte chemoattractant protein-1; mdx, mouse model of DMD; MyD88,Myeloid differentiation primary response 88;NO, Nitric oxide; NOX2, NADPH oxidase 2; PGC-1α, Peroxisome proliferator-activated receptor gamma coactivator 1-alpha; PMSB8, Proteome-derived antimicrobial peptide 8; PC: LysoPC, phosphatidylcholine: lysosome-phosphatidylcholines;PTX3, Pentraxin 3; ROS, Reactive oxygen species; *Tjp* 1 and 2, tight junction proteins; TGF-β, Transforming growth factor beta; Th1, T helper 1; TNF-α, Tumor necrosis factor alpha; TRAF6, Tumor necrosis factor receptor-associated factor 6; Treg, Regulatory T cells; TLR4, Toll-like receptor 4.

**Table 1 ijms-25-11310-t001:** Microbial metabolites and their effects.

Microbial Metabolite	Systemic Effects	Metabolic Effects	Ref.
Short-chain fatty acids (SCFAs)	Anti-inflammatory	Can improve insulin sensitivity, glucose uptake Gut barrier permeability protection Promote ROS formation and mitochondrial energy metabolism Prevent muscle atrophy	[8,9,10] [12,13] [14] [15,16] [17]
Butyrate	Anti-inflammatory	Improves gut barrier function, reduces inflammation, promotes M2 macrophage polarization, regulates muscle cell Akt/mTOR/Foxo3a pathway Promotes PGC-1α gene expression	[8,9] [18]
Secondary bile acids (SBAs)	Anti-inflammatory	Promote mitochondrial biogenesis through coactivator-1α γ receptor activity activated by PGC-1α deacetylation Influence lipid metabolism can modulate inflammation	[8,9] [19,20] [13]
Low levels of hydrogen sulphide (H_2_S)	Anti-inflammatory	Stabilizes mucus layers, inhibits adherence of the microbiota to the epithelium, prevents invasive pathobionts	[14,15]
Branched chain amino-acids	Anti-inflammatory	Regulate myokines, adipokine and skeletal muscle Metabolism Mitigate exercise-induced muscle damage	[21]
Urolithin A	Anti-inflammatory	Improves mitochondrial fatty acid metabolism Stimulates mitochondrial biogenesis regulating PGC-1α and SIRT1	[22] [23]

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
