# Peer review of "The Gut Microbiota Involvement in the Panorama of Muscular Dystrophy Pathogenesis"

_ijms, 2024, doi:10.3390/ijms252011310_

Round 1
Reviewer 1 Report
Comments and Suggestions for Authors
The manuscript by Russo et al. provides a comprehensive overview of the gut-muscle axis with the limited literature there is in the muscular dystrophy field. The figures are very detailed and assist the review greatly.
I have a few comments for the authors to address:
1. The review focuses predominantly on Duchenne muscular dystrophy and the sections on the pathogenesis need to be intertwined with the microbiota effects to help strengthen the review more. As it stands, it reads a little disjointed
2. The review on type 1 myotonic dystrophy is disproportionately less than the review on Duchenne muscular dystrophy and it is not very clear why. Since the review title states muscular dystrophies, the review needs to be more even
3. Once again, since the review title states muscular dystrophies, why are only two included? I think the review would benefit from either focusing it only on Duchenne muscular dystrophy or including more on other dystrophies
4. The figures are very nice but I think a table would help with the details. A table on which microbiota species are known to affect muscle, have anti-/pro-inflammatory effects and metabolic effects would be useful
Comments on the Quality of English Language
Overall the manuscript is well written and only requires review for grammatical errors
Author Response
|
Thank you very much for taking the time to review this manuscript. Please find the detailed responses below and the corresponding revisions/corrections highlighted/in track changes in the re-submitted files
|
|
Point-by-point response to Comments and Suggestions for Authors |
|
Comments 1: The review focuses predominantly on Duchenne muscular dystrophy and the sections on the pathogenesis need to be intertwined with the microbiota effects to help strengthen the review more. As it stands, it reads a little disjointed |
|
Response 1: Thank you for pointing this out. We agree with this comment. Therefore, some sections have been modified as suggested.
|
|
Comments 2: The review on type 1 myotonic dystrophy is disproportionately less than the review on Duchenne muscular dystrophy and it is not very clear why. Since the review title states muscular dystrophies, the review needs to be more even |
|
Response 2: Most of the literature and much experimental data come from studies on models dystrophin-deficient mdx mice. We have described the mechanisms by which dysbiosis may exacerbate the muscle damage of muscular dystrophies. We have intentionally chosen a generic title to arouse curiosity and encourage studies aimed at other types of muscular dystrophy.
Comments 3: Once again, since the review title states muscular dystrophies, why are only two included? I think the review would benefit from either focusing it only on Duchene muscular dystrophy or including more on other dystrophies Response 3: We fully agree with this criticism but there are no studies on microbiota and other types of muscular dystrophy. We believe it is important to report the only study that describes the composition of the microbiota in patients with DM1 to highlight how dysbiosis can affect the exacerbation of dystrophies and not only DMD
Comments 4: The figures are very nice but I think a table would help with the details. A table on which microbiota species are known to affect muscle, have anti-/pro-inflammatory effects and metabolic effects would be useful Response 4: A table describing microbiota metabolites affecting muscle function have been added. Thank you for your valuable suggestion
|
|
Response to Comments on the Quality of English Language |
|
Point 1: Overall the manuscript is well written and only requires review for grammatical errors |
|
Thank you very much for taking the time to review this manuscript. Please find the detailed responses below and the corresponding revisions/corrections highlighted/in track changes in the re-submitted files
|
|
Point-by-point response to Comments and Suggestions for Authors |
|
Comments 1: The review focuses predominantly on Duchenne muscular dystrophy and the sections on the pathogenesis need to be intertwined with the microbiota effects to help strengthen the review more. As it stands, it reads a little disjointed |
|
Response 1: Thank you for pointing this out. We agree with this comment. Therefore, some sections have been modified as suggested.
|
|
Comments 2: The review on type 1 myotonic dystrophy is disproportionately less than the review on Duchenne muscular dystrophy and it is not very clear why. Since the review title states muscular dystrophies, the review needs to be more even |
|
Response 2: Most of the literature and much experimental data come from studies on models dystrophin-deficient mdx mice. We have described the mechanisms by which dysbiosis may exacerbate the muscle damage of muscular dystrophies. We have intentionally chosen a generic title to arouse curiosity and encourage studies aimed at other types of muscular dystrophy.
Comments 3: Once again, since the review title states muscular dystrophies, why are only two included? I think the review would benefit from either focusing it only on Duchene muscular dystrophy or including more on other dystrophies Response 3: We fully agree with this criticism but there are no studies on microbiota and other types of muscular dystrophy. We believe it is important to report the only study that describes the composition of the microbiota in patients with DM1 to highlight how dysbiosis can affect the exacerbation of dystrophies and not only DMD
Comments 4: The figures are very nice but I think a table would help with the details. A table on which microbiota species are known to affect muscle, have anti-/pro-inflammatory effects and metabolic effects would be useful Response 4: A table describing microbiota metabolites affecting muscle function have been added. Thank you for your valuable suggestion
|
|
Response to Comments on the Quality of English Language |
|
Point 1: Overall the manuscript is well written and only requires review for grammatical errors |
|
Response 1: we reviewed grammatical errors |

Reviewer 2 Report
Comments and Suggestions for Authors
Interesting review on the possible pathophysiological role of gut microbiota in muscular dystrophy and related diseases. I have the following comments:
- The role of gut microbiota in modulating pathophysiological aspects of muscle inflammation and dysfunction should not be overestimated. So, some sentences of the abstract should be modulated to underline that gut microbiota may be just one of the players involved, not the only or most important one.
- The authors seem to overlook the role of diet in shaping gut microbiome composition and function. This issue deserves better discussion in the manuscript. The role of SCFAs in modulating gut mucosal permeability should be also highlighted in section 2.
- Besides pathophysiological mechanisms, the authors may also discuss the role of gut microbiome in modulating muscle function. There is evidence that transplantation of dysbiotic human gut microbiota to germ-free mice results into reduced muscle function, while transplantation of gut microbiota from fit humans to germ-free mice determines improvement in muscle function.
- Did any animal study assess the muscular response to damage, for example after trauma, in germ-free mice and in mice colonized with different types of microbiota?
- Are the putative alterations observed in the fecal microbiome of patients with DMD causally linked to alterations of intestinal motility? If so, the gut microbiota dysbiosis associated with DMD may just represent an epiphenomenon of the disease and not one of the concauses leading to loss of muscle function.
- Section 7 should be more focused on the possible role of gut microbiome alterations in driving altered immune response at the muscle level in DMD. The role of dysbiosis emerges only in the figure, but this assumption is not suppoerted by adequate discussion of data in the text.
- Section 8 should discuss the role of gut microbiota as regulator of oxidative stress at the systemic level.
- Is there any clinical study (conducted in humans) reporting alterations of gut microbiota composition in DMD patients?
- Gut barrier function can be assessed with biomarkers, such as serum zonulin. What are serum zonulin levels in mice and human beings with DMD? Do they correlate with parameters of muscle function?
- Strategies to modulate gut microbiota composition and function in DMD should be discussed in a separate paragraph, not in the conclusions section. Is there any evidence that pre, pro or postbiotics may favorably affect any aspect of the disease? The discussion on urolithin A is very interesting, but the concept of urolithin metabotypes should be introduced.
- How do the authors think that the knowledge on the gut muscle axis in DMD can be translated into clinical practice? Should we recommend patients with DMD to take probiotics? should we prescribe them a diet tailored on the microbiome? should we investigate if fecal microbiota transplantation is an effective strategy to modulate the disease?
Comments on the Quality of English LanguageMinor English language errors are present in the manuscript. I recommend a thorough check, while professional English copyediting may not be necessary
Author Response
|
Thank you very much for taking the time to review this manuscript. Please find the detailed responses below and the corresponding revisions/corrections highlighted/in track changes in the re-submitted files
|
|
Point-by-point response to Comments and Suggestions for Authors |
|
Comments 1: The role of gut microbiota in modulating pathophysiological aspects of muscle inflammation and dysfunction should not be overestimated. So, some sentences of the abstract should be modulated to underline that gut microbiota may be just one of the players involved, not the only or most important one. |
|
Response 1: Thank you for pointing this out. We agree with this comment. Therefore, some sentences in the abstract have been modified as suggested.
|
|
Comments 2: The authors seem to overlook the role of diet in shaping gut microbiome composition and function. This issue deserves better discussion in the manuscript. The role of SCFAs in modulating gut mucosal permeability should be also highlighted in section 2. Response 2: In section 2 we mentioned the role of diet in shaping gut microbiome composition and we highlighted the role of SCFAs in modulating gut mucosal permeability as suggested
Comments 3: Besides pathophysiological mechanisms, the authors may also discuss the role of gut microbiome in modulating muscle function. There is evidence that transplantation of dysbiotic human gut microbiota to germ-free mice results into reduced muscle function, while transplantation of gut microbiota from fit humans to germ-free mice determines improvement in muscle function Response 3: In section 2, pag 3 we reported the evidence demonstrating the effect of the transplantation of uncultured and intact human fecal samples from older adult humans into germ-free mice in the maintenance of muscle strength
Comments 4: Did any animal study assess the muscular response to damage, for example after trauma, in germ-free mice and in mice colonized with different types of microbiota Response 4: In this review we discussed the experimental data from studies on models dystrophin-deficient mdx mice such as the investigations of Lahiri S et al and Jollet M et al. We are not aware of many other studies on this subject.
Comment 5: Are the putative alterations observed in the fecal microbiome of patients with DMD causally linked to alterations of intestinal motility? If so, the gut microbiota dysbiosis associated with DMD may just represent an epiphenomenon of the disease and not one of the concauses leading to loss of muscle function. Response 5: Section 9 pag 12 we followed your suggestion to point out that dysbiosis associated with DMD may just represent an epiphenomenon of the disease and not one of the concauses leading to loss of muscle function.
Comment 6: Section 7 should be more focused on the possible role of gut microbiome alterations in driving altered immune response at the muscle level in DMD. The role of dysbiosis emerges only in the figure, but this assumption is not suppoerted by adequate discussion of data in the text. Response 6: Section 7 has been modified has you suggested.
Comment 7: Section 8 should discuss the role of gut microbiota as regulator of oxidative stress at the systemic level. Response 7: Section 8 we have discussed the role of gut microbiota on oxidative stress at systemic level has you suggested. Thank you.
Comment 8: Is there any clinical study (conducted in humans) reporting alterations of gut microbiota composition in DMD patients? Response 8: To our knowledge, there is no clinical studies reporting alterations of gut microbiota on patients
Comment 9: Gut barrier function can be assessed with biomarkers, such as serum zonulin. What are serum zonulin levels in mice and human beings with DMD? Do they correlate with parameters of muscle function? Response 9: In section 12 we discussed on the use of zonulin as a potential marker as suggested by you
Comment 10: Strategies to modulate gut microbiota composition and function in DMD should be discussed in a separate paragraph, not in the conclusions section. Is there any evidence that pre, pro or postbiotics may favorably affect any aspect of the disease? The discussion on urolithin A is very interesting, but the concept of urolithin metabotypes should be introduced. Response 10: Following your suggestion, paragraph 12 was divided into two separate sections. In the section 13 we have introduced the concept of urolithin metabotypes
Comment 11: How do the authors think that the knowledge on the gut muscle axis in DMD can be translated into clinical practice? Should we recommend patients with DMD to take probiotics? should we prescribe them a diet tailored on the microbiome? should we investigate if fecal microbiota transplantation is an effective strategy to modulate the disease? Response 11: As commented in the conclusion to all questions of this concern it will be possible to answer only when clinical studies on patients with muscular dystrophy will carried out |

Round 2
Reviewer 1 Report
Comments and Suggestions for Authors
Dear authors,
Thank you for addressing my feedback. I appreciate the effort you have put in and believe the manuscript has been improved.
Author Response
Thank you very much for taking the time to review this manuscript. ank you very much for taking the time to review this manuscript.
Reviewer 2 Report
Comments and Suggestions for Authors
The authors have responded adequately to all my previous questions.
Just few minor issues:
- In Table 1, butyrate should be included in the same row as short chain fatty acids, because it belongs to this category.
- In the conclusions section, I advice authors not to formulate areas of uncertainty for future research in the form of question. The second-last paragraph should be rephrased.
- "Data availability statement" does not refer to the search strategy for the present review article. It is a statement generally required for articles reporting original data. I think the statement should be changed.
Author Response
|
Thank you very much for taking the time to review this manuscript. Please find the detailed responses below and the corresponding revisions/corrections highlighted/in track changes in the re-submitted files
|
|
Point-by-point response to Comments and Suggestions for Authors |
|
Comments 1: In Table 1, butyrate should be included in the same row as short chain fatty acids, because it belongs to this category. |
|
Response 1: The table has been correct
|
|
Comments 2: - In the conclusions section, I advise authors not to formulate areas of uncertainty for future research in the form of question. The second-last paragraph should be rephrased. Response 2: In the conclusions section the paragraph has been reworded as you suggested
Comments 3: "Data availability statement" does not refer to the search strategy for the present review article. It is a statement generally required for articles reporting original data. I think the statement should be changed.
Response 3: The data availability statement has been deleted
|